# tracerDB: a crowdsourced fluorescent tracer database for target engagement analysis

Johannes Dopfer [1,2], James D. Vasta[3], Susanne Müller [1,2], Stefan Knapp[1,2,4], Matthew B. Robers [3] & Martin P. Schwalm [1,2,4] ✉

Investigating ligand-protein complexes is essential in the areas of chemical biology and drug discovery. However, detailed information on key reagents such as fluorescent tracers and associated data for the development of widely used bioluminescence resonance energy transfer (BRET) assays including NanoBRET, time-resolved Förster resonance energy transfer (TR-FRET) and fluorescence polarization (FP) assays are not easily accessible to the research community. We created tracerDB, a curated database of validated tracers. This resource provides an open access knowledge base and a unified system for tracer and assay validation. The database is freely available at https://www.tracerdb.org/.

Well-characterized, selective small molecules—"chemical probes"—are essential tools for target validation during drug development and in basic biological research[1]. Criteria for small molecule modulators to qualify as chemical probes have been established by chemical biologists and are widely accepted in the community[2]. These include target-related criteria for potency, selectivity, and proof for target engagement in addition to the suitability of the chemical matter itself[1]. By creating these quality criteria, chemical probes became important and generally recognized tools aiding the scientific community and accelerating drug discovery. Inspired by this approach, our goal is the standardization of quality criteria within the drug candidate evaluation process. For the evaluation of a ligand-protein interaction, either direct or indirect measurements can be carried out. Direct binding assays such as isothermal titration calorimetry (ITC) and surface plasmon resonance (SPR) are state-of-the-art methods to measure dissociation constants ($K_D$). Additionally, ITC measurements are both label-free and do not require immobilization. They allow for the determination of the stoichiometry and thermodynamic parameters, whereas SPR enables the determination of the binding rate constants such as $k_{on}$ and $k_{off}$. However, these methods either require large amounts of protein (ITC) or immobilized purified protein (SPR) and therefore are excellent tools for final binding validation but not optimal for larger screening and in-cell campaigns. Indirect biochemical and cellular assays often rely on in-solution (or in-cell) displacement assays using fluorescence-labeled

molecules, called tracers (sometimes referred to as fluorescent probes—not to be confused with chemical probes themselves or medical radiotracers)[3–5]. Tracers are composed of (1) a moiety that binds to the protein of interest (POI), such as small molecules, DNA, RNA, and peptides, (2) a chemical linker, and (3) a reporter label, typically a fluorescent dye[6,7]. To avoid interference of the linker with the binding of the molecule to the POI, the choice of the right exit vector, a solvent exposed attachment point of the linker to the molecule, is important (Fig. 1a).

Tracers are used in cellular target engagement assays (in cellulo) such as time-resolved Förster resonance energy transfer (TR-FRET)[6] or bioluminescence resonance energy transfer (BRET)[4] assays or biochemical in vitro studies, which can be BRET-based, TR-FRET-based or comprise fluorescence polarization (FP)[8]. In particular, NanoBRET, a method frequently applied in kinase live-cell target engagement assays, critically relies on the use of suitable tracer molecules. This method validates the binding of a small molecule such as an inhibitor to its cognate target in the cell. It is also suitable for assessing cellular selectivity by utilizing a single tracer[9]. Owing to the stringent distance and orientation constraints of the BRET donor, tracers do not have to be specific for the protein of interest. Promiscuous BRET tracers are ideal as they survey multiple targets. Using this principle, we successfully enabled 206 (as of Feb. 2024) validated kinase interactions with tracer K10 (T000008).

[1]Institute of Pharmaceutical Chemistry, Goethe University Frankfurt, Max-von-Laue-Str. 9, 60438 Frankfurt am Main, Germany. [2]Structural Genomics Consortium, Goethe University Frankfurt, Buchmann Institute for Life Sciences, Max-von-Laue-Str. 15, 60438 Frankfurt am Main, Germany. [3]Promega Corporation, Madison, WI, USA. [4]German Cancer Consortium (DKTK)/German Cancer Research Center (DKFZ), DKTK Site Frankfurt-Mainz, 69120 Heidelberg, Germany. ✉e-mail: schwalm@pharmchem.uni-frankfurt.de

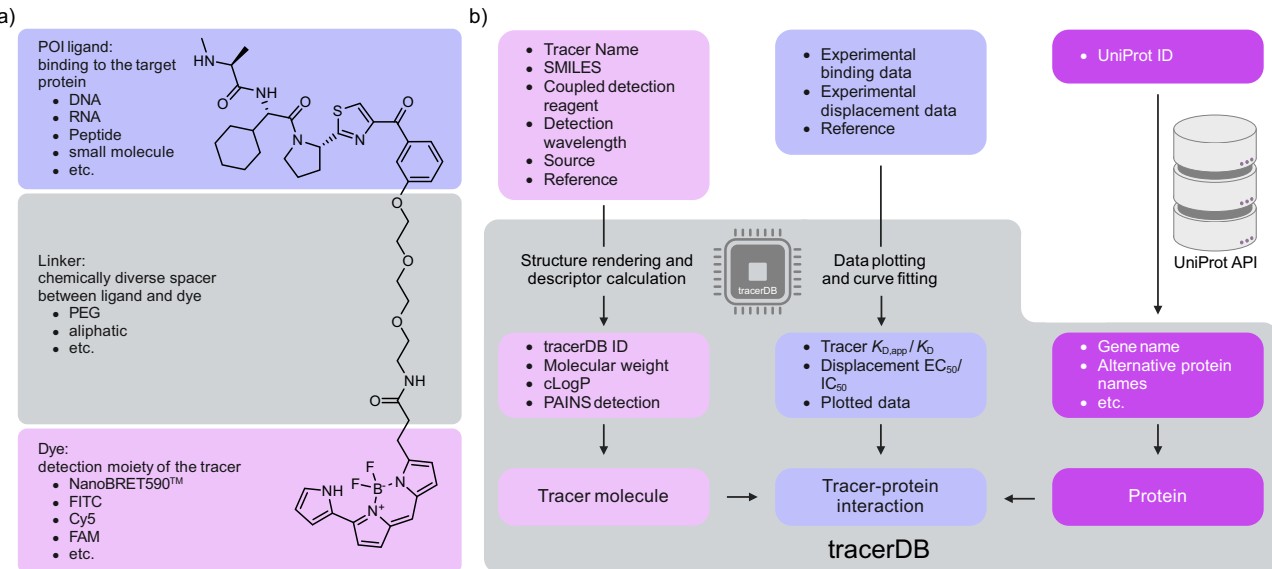

**Fig. 1 | Composition of a tracer (T000001)[22] and the principle underlying the tracerDB. a** Schematic representation of a tracer molecule with its three distinct substructures. The POI ligand marks the binding moiety to the target protein, ultimately generating the proximity between target and the label. The optical reporter (dye) is chemically linked to this POI ligand via a chemical linker. The attached dye is selected according to the requirements of the assay system to achieve the desired excitation and emission wavelengths. **b** Underlying principles of tracer and experimental data processing. Information provided through submission is displayed in the upper panels outside the tracerDB framework. These data are parsed and assigned to each entity: the tracer molecule, the protein and interaction between the former as shown in detail in Fig. 2.

## Results

Due to the importance of the quantification of protein-ligand interactions, a large number of tracers are reported within the literature. However, scientists face several problems to establish displacement assays for their respective target: (1) finding established tracers in the literature using search engines is difficult, as much of the required information is buried in the Supplementary Methods; (2) reproducibility of the reported assays is often problematic due to insufficient validation of the tracer or unfavorable assay parameters; (3) the availability of the tracer is often unknown. We created a database for fluorescent tracer molecules named tracerDB to address these problems. It has been developed and standardized to provide design and application guidance based on strict performance criteria. For each tracer-based assay, the chemical structure or commercial availability is provided, as well as the assay parameters and a reference. tracerDB allows to search for the protein of interest or the tracer, enabling fast assessment of available assay options for a specific target. Within the first 6 months (as of April 2024), 42 tracers, targeting 318 different proteins in 476 experimentally validated assays were reviewed and uploaded.

Scientists worldwide can submit their tracer data for review and inclusion in the database. The submission of tracer data must contain all necessary information (no physical molecules) required to judge the quality and reproducibility of a tracer-based assay. First, general information about the molecular structure (e.g. simplified molecular-input line-entry system (SMILES) specification, fluorophore characteristics, storage conditions and trivial name) are required for the creation of a tracer page (Fig. 1b). In some special cases, tracer structures cannot be disclosed. In this case, the availability of the tracer must be guaranteed to allow access to all reported assays which would otherwise be granted by the chemical structure of the tracer. Every structure submitted is checked for structural features associated with "pan assay interference compounds" (PAINS)[10] which are reported together with the tracer information. Since the applied filters for the detection of PAINS also detect fluorescent substructures, it is recommended to inspect the highlighted moieties of the tracers that are flagged, by opening the PAINS report within the tracer description

panel. All target proteins bound by the tracer have to be listed in UniProt[11]. Experimental data for the tracer titration and compound displacement are part of the validation process and must be uploaded together with information on a recommended concentration, the $Z'$ value of the assay, and the assay window observed. Here, the assay window describes the fold-change between signal (tracer bound) and noise (tracer only) at the recommended tracer concentration (https://www.tracerdb.org/about/). The experimental data are available for download by the user. To facilitate the upload and review process, data can be submitted via the submission page (https://www.tracerdb.org/submission). On this page, all information can be added with essential information written in bold. Without providing all necessary information, the submission is not possible. After insertion of all required information, the data is automatically sent to info@tracerdb.org, for final approval and upload, allowing submission without the need of a login by the user. Additionally, tracer IDs can be assigned prior to publication, allowing a direct link to the database (in analogy to PDB). In contrast to an automatically generated data repository, the submission of tracer data is followed by a review process which makes tracerDB a reviewed and curated database. Thus, every entry has been examined for its agreement with the database's quality criteria, allowing adherence to the highest possible quality control.

The interaction network between tracer molecules and their respective targets can be modeled as a many-to-many relationship where many tracers can bind a single protein and a single tracer can bind many proteins. As a result, the underlying database structure consists of three entity sets: the tracer, the protein, and their interaction (Fig. 1b). To ensure a user-friendly submission of data and standardize the presentation, all molecular representations and calculations are created and executed on the server side. We chose Django[12] as a python-based web framework together with a MySQL database to enable high-frequency read operations.

In addition to the information on the crowdsourced tracers, we have also included general information on tracer molecules and illustrations of different assay systems on the "about" page (https://www.tracerdb.org/about/). Here, we describe the quality control criteria and how to calculate the respective values. In order to further increase the

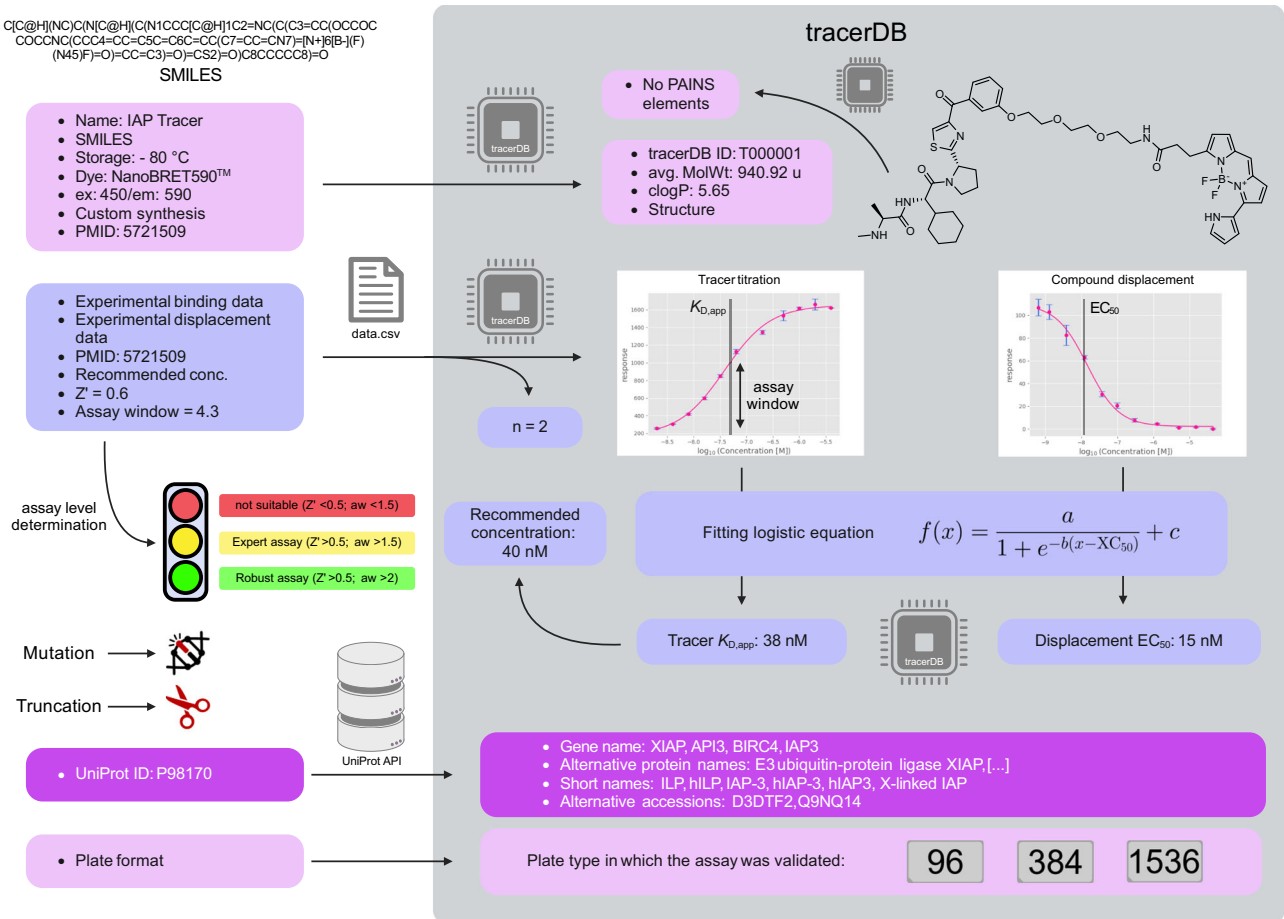

**Fig. 2 | Data input and processing carried out by the webserver.** Input data are depicted on the left, outside the tracerDB framework. Calculations carried out by the database are marked with a processor symbol. First, the database calculates tracer parameters and generates a schematic representation of the tracer molecule including an automated detection of PAINS elements within the tracer structure. Next, assay parameters and experimental data are uploaded and processed. From the experimental data (.csv file) the number of replicates is extracted and the datapoints are plotted. The data are interpolated using the indicated function to yield the tracer $K_D$ and displacement $IC_{50}/EC_{50}$. The recommended tracer concentration is estimated from the tracer $K_D$, but can be changed if more optimal conditions are known. Finally, the target is registered using its UniProt ID, resulting in searchable accession numbers, gene, and protein names.

reproducibility of the described assays, each assay is classified according to its parameters into robust, expert and unsuitable assays with exemplary data for clarification (Fig. 2). These assay levels are represented by a traffic light icon for each registered assay. In addition, we have included a methods section describing the different assays used to collect the submitted data (https://www.tracerdb.org/methods/). This section is supported by an illustration and key references.

## Discussion

The development and implementation of tracer-based assays is carried out by countless laboratories around the world. For every assay, proper validation and standardization are crucial to ensure assay quality. To support the reproducibility of established assays across different laboratories, tracerDB helps to standardize assays by providing a curated and constantly growing set of experimentally validated tracers with recommended concentrations. Additionally, the tracerDB "about" page (https://www.tracerdb.org/about/) summarizes the most important quality control information to ensure the generation of high-quality data. Further information on validated exit vectors for the development of other bifunctional compounds or indications of suitable protein fusion termini can be extracted from the database, as well.

tracerDB is therefore a resource for drug-screening scientists as well as the chemical biology community, that gathers detailed, reviewed and high-quality information on tracer-based assays and their applications.

## Methods

### Architecture of the database

RDkit[13], a commonly used cheminformatics package for python is employed to render SMILES strings as two-dimensional molecular representations, along with the implemented substructure search to allow for the detection of PAINS elements and their depiction. The average molecular weight and the estimated log$P$ value of the compound- and peptide-based tracers are calculated using RDkit's implemented methods for molecular descriptors. In order to avoid having to deal with complex SMILES of large peptide tracers, the pyPept package[14] has been incorporated into this project to allow for flexible declaration of custom amino acids, i.e. fluorophore peptide labels. These artificial building blocks are then included into the string representation of the peptides and stored in the database as BILN[15]. For the interactive depiction of the three-dimensional structure of protein-based tracers, the NGL viewer was incorporated[16,17]. To ensure consistency in the depiction and analysis of experimental data uploaded to the webserver, fitting and plotting are executed on the server side. The

experimental titration data is plotted via Matplotlib[18] and the fitting is conducted through SciPy[19] using non-linear least squares optimization. It is assumed that the data from concentration response experiments exhibit a sigmoidal shape. Hence, to fit the data the following logistic equation is employed:

$$f(x) = \frac{a}{1 + e^{-b(x - \mathrm{XC}_{50})}} + c \qquad (1)$$

The response of the measurement is a function of the logarithmic concentration $x$, with the additional parameters $a$, $b$, and, $c$ which are utilized to scale and transform $f$, because the input is not normalized. $\mathrm{XC}_{50}$ is the parameter determining the log concentration halfway between the plateaus of the sigmoidal curve. Depending on the experimental context this parameter may be interpreted as $\mathrm{EC}_{50}$ or $\mathrm{IC}_{50}$. Protein titrations performed during the development of fluorescence polarization assay are commonly plotted as signal in milli-polarization units versus the molar concentration. These saturation curves are estimated using the following hyperbolic model:

$$f(x) = \frac{B_{\max} \times x}{K_D + x} + cx + d \qquad (2)$$

where $B_{\max}$ denotes the extrapolated maximum specific binding to the protein for high ligand concentrations. $K_D$ is the equilibrium dissociation constant, which specifies the concentration $x$ required for half-maximum binding at equilibrium. The parameter $c$ accounts for the ratio of nonspecific binding to total binding and $d$ corrects for background signals[20].

Protein information is automatically retrieved through the Uni-Prot REST API, enabling the search for alternative protein and gene names. The retrieved XML files are processed using Biopython's Uni-Prot parser[21], resulting in standardized and well-annotated protein entries, ultimately leading to more robust search functionality.

**Reporting summary**

Further information on research design is available in the Nature Portfolio Reporting Summary linked to this article.

## Data availability

All data are available as download within the database.

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

## Acknowledgements

The authors are thankful for all current and future tracer submissions from diverse laboratories, especially the extensive submissions of the Arrowsmith and Mazitschek Labs. M.P.S., J.D., S.M. and S.K. are grateful for support by the Structural Genomics Consortium (SGC), a registered charity (no: 1097737) that receives funds from Bayer AG, Boehringer Ingelheim, Bristol Myers Squibb, Genentech, Genome Canada through Ontario Genomics Institute, EU/EFPIA/OICR/McGill/KTH/Diamond Innovative Medicines Initiative 2 Joint Undertaking [EUbOPEN grant 875510], Janssen, Merck KGaA, Pfizer and Takeda, and by the German Cancer Research Center DKTK, and the Frankfurt Cancer Institute (FCI). M.P.S. is funded by the Deutsche Forschungsgemeinschaft (DFG, German Research Foundation), CRC1430 (Project-ID 424228829). J.D.V. and M.B.R. are employees of Promega Corp.

## Author contributions

J.D.: website development, data processing and manuscript editing. J.D.V., S.M., S.K. and M.B.R.: manuscript editing. M.P.S.: conceptualization, data processing, website editing, manuscript preparation and editing.

## Funding

## Competing interests

J.D.V. and M.B.R. are employees of Promega. The remaining authors declare no competing interests.
