## [Peer Review File · Nature Communications]

tracerDB: A crowdsourced fluorescent tracer database for target engagement analysisREVIEWER COMMENTS

Reviewer #1 (Remarks to the Author):

This manuscript by Dopfer et al. describes ongoing efforts to create a crowdsourced database for target engagement analysis. The motivation to provide a one-stop place for information on the characteristics and availability of tracer molecules is laudable and could be helpful. Currently, the database contains some 42 tracers, which are reported to cover 318 proteins. However, it should be noted that one tracer, K10, accounts for 206 of the proteins, and many of the entries appear to be sold by Promega, which has a website describing the probes. It is too early to say whether this database will be adopted widely by the scientific community. Given the voluntary nature of the data deposition, it is difficult to see the database being comprehensive. So, a diligent scientist will still search the primary literature for alternatives. However, it does represent a good starting point, and it also serves to set standards regarding the robustness of a tracer assay. I believe the database has potential, but I am skeptical of the motivation of investigators to submit tracer information to the database. I would be more confident if there were a more proactive effort to populate the database.

Reviewer #2 (Remarks to the Author):

This manuscript details conceptualisation, collation, and review of fluorescent tracers to develop “tracerDB”, an open-access, unified, database that enables researchers to easily identify validated tracers for target proteins.

Whilst this database will no doubt prove useful for researchers in the chemical biology or drug discovery field, there remain some points to address before publication, namely the completeness of the data curated on the database, and the transparency of the submission and review process of tracers submitted to tracerDB. Given the specialised nature of this resource, lack of novel data/insights, and limited impact for researchers not using or developing these types of tracers, submission to a more specialised journal such as Communications Biology or Communications Chemistry may be more appropriate.

Major points

Whilst the database gives assay information on recommended tracer concentration, assay window and Z' score, it would benefit greatly from encompassing a wider breadth of assay information, for example: protein concentration, buffer composition, assay volume, incubation periods, temperatures, DMSO tolerance, plate format (96, 384, 1536-wells), plate reader settings etc. Brief analysis on compound suitability (eg PAINs motifs) would also be beneficial. Having all this

information unified for each tracer and protein combination would increase the impact of the database and further decrease the time spent searching for this information by researchers. Additional detail in the manuscript as to the parameters included in the database would also be useful for reader clarity.

More information is required on how the review of tracer submissions is handled. Will the submitted tracers be critically reviewed by academics, or will they be uploaded to tracerDB as long as the required information is submitted? The abstract figure indicates industrial partners; more details on these partners and their role in the database should be included on both the website and briefly in the manuscript. The incorporation of a review process is a very important distinction between a curated database and a repository of data, and therefore this is vital information to include in the manuscript.

Parameters outlining tracer stability (storage and photobleaching) and solubility would be a very useful feature for researchers using the database.

Minor points

In Figure 2 the assay window is shown as 50% maximum signal, however on the tracerDB “about” section the assay window is defined as “the fold-change of saturated tracer-bound signal in contrast to free protein”. Please ensure consistent definitions.

A list of references for a tracer when it has been used in the literature would be beneficial for users.

Filter searching based on the type of assay. For example, it may be useful for researchers to specifically search for FP-assay probes – this may help then compare assay conditions, tracer structures and design novel tracers.

Reword first sentence of the abstract – investigation is not a technology.

Page 4, line 77 should read “prior to publication”.

Page 4, line 79; the sentence describing the interaction network of tracers as a ‘many-to-many relationship’ requires additional clarification.

Remove the paragraph about PROTAC design as it is not relevant nor supported by evidence in the manuscript.

Reviewer #3 (Remarks to the Author):

The manuscript by Dopfer et al. reports on the development of TracerDB, a publically accessible online database for fluorescent tracer ligands for homogenous and cellular target engagement

assays based on optical readout methods that cover the most frequently used methodologies BRET, TR-FRET and FP.

TracerDB fills an unmet need and I am confident that the community will embrace it. The fact that the portal is housed within the framework of the Structural Genomics Consortium (SGC) is comforting as it will likely ensure stability and continued support, which unfortunately has not been the case for many other online databases that had promising starts but depended on the maintenance by individual labs. The implementation is clean and easy to use, and the accompanying manuscript provides a good overview. However, before I can recommend the manuscript for acceptance, there are a couple of concerns that should be addressed, as well as some suggestions the authors may want to consider.

- It would be helpful if the authors would provide a critical consideration of alternative methodological approaches, such as ITC and SPR, which are commonly used and do not rely on fluorescent tracer ligands.

- In line 68 the authors state that the chemical structure of the tracer ligand is required upon submission. However, for the K10 tracer (T000008) that is introduced above as an example, the structural information appears to be absent.

- The authors also state that a tracer that is non-selective is “ideal” (line 48). While this may allow for using a tracer for multiple targets, there are other limitations when employing the tracer in a cellular context, as binding to other targets will deplete the tracer and may have immediate consequences on the cellular response by inhibiting many kinases at the same time.

- It would be desirable if the raw data that had been submitted for a given tracer were accessible to users.

- Providing a general best practice approach, such as reported in this paper PMID: 32758356, would be helpful to ensure high-quality standards.

Reviewer #1 (Remarks to the Author):

This manuscript by Dopfer et al. describes ongoing efforts to create a crowdsourced database for target engagement analysis. The motivation to provide a one-stop place for information on the characteristics and availability of tracer molecules is laudable and could be helpful. Currently, the database contains some 42 tracers, which are reported to cover 318 proteins. However, it should be noted that one tracer, K10, accounts for 206 of the proteins, and many of the entries appear to be sold by Promega, which has a website describing the probes. It is too early to say whether this database will be adopted widely by the scientific community. Given the voluntary nature of the data deposition, it is difficult to see the database being comprehensive. So, a diligent scientist will still search the primary literature for alternatives. However, it does represent a good starting point, and it also serves to set standards regarding the robustness of a tracer assay. I believe the database has potential, but I am skeptical of the motivation of investigators to submit tracer information to the database. I would be more confident if there were a more proactive effort to populate the database.

We thank the reviewer for comments highlighting the value of our database for academic research. We agree with the reviewer that creating and maintaining a database from scratch has some inherent hurdles, especially in a crowdsourced form. We still believe that a crowdsourced, reviewed and experimental data driven database is much more useful for the scientific community than a repository due to the reduced risk of wrong annotations, multiplications or the lack of quality criteria during automated data fetching campaigns. Additionally, many tracers are only briefly mentioned in journal methods often without disclosure of their properties and key validation experiments. Hence, a submission of these molecules to the database would make the method annotation more reproducible and meaningful.

To initiate a crowdsourced database, we had to populate it to a certain extent to highlight the potential and motivate laboratories to submit their data. To quickly gather a broad range of data, we were supported by Promega (NanoBRET tracers), the Arrowsmith lab from University of Toronto (FP tracers) and the Mazitschek lab (TR-FRET tracers) from Harvard Medical School. A trade-off was made by incorporating undisclosed tracers from Promega without a chemical structure but guaranteed accessibility of this tracer for every laboratory. Since many laboratories cannot synthesize the molecules themselves, we found guaranteed availability equally important as a chemical structure. We therefore added this to the submission criteria. Additionally, despite the tracers are described on Promega's website, there is no such curated list as available within the database. Moreover, we include non-catalogue assays which are not reported by Promega.

We are currently actively promoting the database at scientific conferences as within 2024 the SLAS in Boston, the TPD Summit in London or the AACR in San Diego. We are convinced that a publication in a highly visible journal such as Nature Communications will significantly increase the awareness and the support for the database. Additionally, many laboratories already assured the submission of their tracers which we expect in the near future.

We believe that the database (together with a peer reviewed publication) might harbour the potential to increase quality control and availability within the literature as it can be seen for the PDB, which is a crowdsourced database, carrying out experimental structure validation for scientists and journals while making all structures publicly available. To further increase awareness on quality criteria during assay development, we implemented an additional paragraph to the 'about' page.

We agree with the reviewer that a single publication will not be sufficient to make the database sustainable and we will continue to actively approach different laboratories publishing tracers to encourage them to submit their data to the database. This way, we will also be able to make the database more comprehensive.

Reviewer #2 (Remarks to the Author):

This manuscript details conceptualisation, collation, and review of fluorescent tracers to develop “tracerDB”, an open-access, unified, database that enables researchers to easily identify validated tracers for target proteins.

Whilst this database will no doubt prove useful for researchers in the chemical biology or drug discovery field, there remain some points to address before publication, namely the completeness of the data curated on the database, and the transparency of the submission and review process of tracers submitted to tracerDB. Given the specialised nature of this resource, lack of novel data/insights, and limited impact for researchers not using or developing these types of tracers, submission to a more specialised journal such as Communications Biology or Communications Chemistry may be more appropriate.

We thank the reviewer for these comments. Since the database depends on a crowd-sourcing approach, we submitted this manuscript to a journal with a broad readership to reach a diverse audience. Publication in a biology or chemistry focused journal would only reach a small part of the community we want to engage in this initiative. The database, on the other hand, requires input from the chemistry/chemical biology community, but is aimed at the broader biological user community. Therefore, we strongly believe that a journal with a broader audience is best suited to launch and sustain this initiative.

Major points

Whilst the database gives assay information on recommended tracer concentration, assay window and Z' score, it would benefit greatly from encompassing a wider breadth of assay information, for example: protein concentration, buffer composition, assay volume, incubation periods, temperatures, DMSO tolerance, plate format (96, 384, 1536-wells), plate reader settings etc. Brief analysis on compound suitability (eg PAINS motifs) would also be beneficial. Having all this information unified for each tracer and protein combination would increase the impact of the database and further decrease the time spent searching for this information by researchers. Additional detail in the manuscript as to the parameters included in the database would also be useful for reader clarity.

The reviewer pointed out important parameters which will notably improve the usability of the database.

We have now included the following changes:

- *We added references to the about page including an explanation on the binding and titration regime, reported in PMID: 32758356. Here, the authors highlight the importance of assay validation and we try to raise awareness of possible errors during the measurement of binding in the about page.*
- *We now link the plate format to the assay in form of new icons*
- *Recommended storage temperature is now included in the tracer page*
- *We included a PAINS filter for all submitted structures with a corresponding statement printed on the tracer page*

- *Submission is now changed to a submission page instead of an excel file. Within this process we encourage for the submission of additional information such as protein concentration, buffer conditions and constructs used*
- *We now included a function which allows the download of the underlying experimental data*

More information is required on how the review of tracer submissions is handled. Will the submitted tracers be critically reviewed by academics, or will they be uploaded to tracerDB as long as the required information is submitted?

Currently, reviewing is performed by the authors of this manuscript. We however intend to establish a more diverse review board in the future. During the review process, we check the submitted data with the help of an underlying publication. We will also re-plot the data and compare the graphs with the peer-reviewed graphs if available in the original publication. In addition, we review the method section for scientific soundness and extract information such as antibody concentrations (as used in TR-FRET) from corresponding method sections. This information will be included in the comment section of the assay data page. Lastly, we ensure that the submitted data matches our quality criteria, before data are uploaded into the database.

As pointed out by reviewer #1, we now also included additional required information in the submission form (now changed to a submission page) which can be used to cross-check the reported methods. If the database develops as we hope, we will nominate an independent reviewing committee to ensure quality of the data (as it was already done for the Probes Portal).

The abstract figure indicates industrial partners; more details on these partners and their role in the database should be included on both the website and briefly in the manuscript.

The role of industrial partners will be the same as for academic partners. Tracers and their profiling data are submitted to the tracer database. While most tracers from academia stem from published data, some data from industry, originate from internal efforts and the database marks an opportunity to present so far unpublished data. Due to the lack of graphical abstracts in this journal, the figure will be removed.

The incorporation of a review process is a very important distinction between a curated database and a repository of data, and therefore this is vital information to include in the manuscript.

We agree with the reviewer and we made this important point now clearer in the revised manuscript.

Parameters outlining tracer stability (storage and photobleaching) and solubility would be a very useful feature for researchers using the database.

We agree that storage and other parameters are very useful to the user community and we have added these parameters now to the tracer information page.

Minor points

In Figure 2 the assay window is shown as 50% maximum signal, however on the tracerDB “about” section the assay window is defined as “the fold-change of saturated tracer-bound signal in contrast to free protein”. Please ensure consistent definitions.

We apologise for the oversight in amending the about section and changed the statement accordingly.

A list of references for a tracer when it has been used in the literature would be beneficial for users.

We agree that this might be useful, and we will try to implement this in a future version. We are currently working on such a system, but unfortunately, we have not yet identified a suitable and stable solution for the automated identification of underlying publications. We will continue to work on this topic in the future.

Filter searching based on the type of assay. For example, it may be useful for researchers to specifically search for FP-assay probes – this may help then compare assay conditions, tracer structures and design novel tracers.

We have included a filtered search which can be used through the advanced search options in the main search bar.

Reword first sentence of the abstract – investigation is not a technology. Page 4, line 77 should read “prior to publication”. Page 4, line 79; the sentence describing the interaction network of tracers as a ‘many-to-many relationship’ requires additional clarification.

We thank the reviewer for spotting this error and we corrected it as suggested.

Remove the paragraph about PROTAC design as it is not relevant nor supported by evidence in the manuscript.

We agree with the reviewer that this statement is not relevant for the manuscript. We therefore excluded this paragraph and changed the statement on the ‘about’ page accordingly by adding a reference, where Palbociclib-FITC (T000036) was synthesized in analogy to cereblon degraders.

Reviewer #3 (Remarks to the Author):

The manuscript by Dopfer et al. reports on the development of TracerDB, a publically accessible online database for fluorescent tracer ligands for homogenous and cellular target engagement assays based on optical readout methods that cover the most frequently used methodologies BRET, TR-FRET and FP.

TracerDB fills an unmet need and I am confident that the community will embrace it. The fact that the portal is housed within the framework of the Structural Genomics Consortium (SGC) is comforting as it will likely ensure stability and continued support, which unfortunately has not been the case for many other online databases that had promising starts but depended on the maintenance by individual labs. The implementation is clean and easy to use, and the accompanying manuscript provides a good overview. However, before I can recommend the manuscript for acceptance, there are a couple of concerns that should be addressed, as well as some suggestions the authors may want to consider.

We thank the reviewer for the support of the project.

- It would be helpful if the authors would provide a critical consideration of alternative methodological approaches, such as ITC and SPR, which are commonly used and do not rely on fluorescent tracer ligands.

We now included such consideration in the manuscript as well as in the "about page".

- In line 68 the authors state that the chemical structure of the tracer ligand is required upon submission. However, for the K10 tracer (T000008) that is introduced above as an example, the structural information appears to be absent.

Indeed, the structure for some tracers are undisclosed. As also replied to reviewer #1 who had a similar comment, we were required to populate the database as a motivation for laboratories to use it and ultimately submit their own data. During the initial phase, we also included undisclosed tracers. However, with the exception of the structure, all associated data necessary to assess the quality of the assays are provided.

A main motivation was to ensure availability of the tracers, which is also one of the reasons, why we generally ask for a structure and a synthesis reference. If accessibility is ensured, for instance by a commercial source, it is anticipated that the data will still be of benefit to the scientific community in the future.

We included this consideration in the manuscript that undisclosed tracers are only suitable, if accessibility by the submitting laboratory is guaranteed. Based on these criteria, we are able to include tracers that would not normally be in the public domain.

- The authors also state that a tracer that is non-selective is "ideal" (line 48). While this may allow for using a tracer for multiple targets, there are other limitations when employing the tracer in a cellular context, as binding to other targets will deplete the tracer and may have immediate consequences on the cellular response by inhibiting many kinases at the same time.

We agree with the reviewer that this statement requires more clarification. Due to this, we included a paragraph on this topic at the about page together with experimental data to demonstrate the usability of promiscuous ligands.

“Some of the reported tracers contain highly promiscuous target ligands. This approach is possible since either isolated systems (FP and TR-FRET) or proximity-based methods (NanoBRET and TR-FRET) are used. The latter only allows the detection of tracer in proximity to the NLuc- or Lanthanide-tagged protein and therefore disregards the unbound fraction which might interact with additional targets in case of cellular systems. In case of cell-based measurements and therefore a system with a number of additional binding partners, the ratio of culture medium to cell volume is $\sim 10^4$ and together with the given cell permeability, we can assume a system with $[\text{tracer}]_{\text{total}} \approx [\text{tracer}]_{\text{free}}$. To test this hypothesis, a comparison of a highly promiscuous tracer (T00008) and a notably more selective tracer based on Palbociclib are compared below in a NanoBRET assay for CDK4. Ensuring an assay which is measured in binding regime and in agreement with the Cheng-Prusoff equation, the selective tracer displayed approx. 6-fold higher affinity towards CDK4 and was ultimately used in 8-fold higher concentration. Nevertheless, both displacement curves using Palbociclib were found to be in high agreement, proving the promiscuous tracer approach suitable for tracer development.”

- It would be desirable if the raw data that had been submitted for a given tracer were accessible to users.

We agree with the reviewer and implemented a download function for the data.

- Providing a general best practice approach, such as reported in this paper PMID: 32758356, would be helpful to ensure high-quality standards.

We now include a part in our about page to raise awareness on the potential problem to measure in the titration regime with an equilibrated system.

“The use of too high protein/tracer concentrations can lead to apparently weaker affinities due to tracer competition. To avoid this, users need to ensure that data were measured in the “binding regime” according to Jarmoskaite et al. The binding regime requires an assay at which the constant component is well below the KD (\ll KD) which is e.g. given by the low concentration of NanoLuciferase-tagged protein expressed by the cells, low protein concentrations in TR-FRET or low tracer concentrations in FP assays. The use of concentrations that are too high will result in an assay being in the “titration regime”, a system that is likely to produce artefacts. Additionally, it is recommended to ensure an equilibrated system to avoid similar artefacts. Therefore, before developing tracers and setting up binding assays, we recommend that the system is tested to be in the “binding regime” and equilibrated as described by the authors.”

REVIEWERS' COMMENTS

Reviewer #2 (Remarks to the Author):

The authors have made significant efforts to address the concerns of the previous review, and the authors' references to following the format of the Chemical Probes Portal provides confidence that the database will be appropriately managed following increased user engagement. Additionally, the authors' statement regarding publishing in a less specialised journal (instead of a chemistry or biology-focussed journal) to gain access to a wider audience is valid. We would thus recommend for publication in Nature Communications following minor revisions.

Minor points

The concern regarding assay window definition has not been adequately addressed, with differing/conflicting definitions on tracerDB and the manuscript. In the manuscript the term 'assay window' is not defined and the figure shows 50% max signal, whilst on the tracerDB 'about' section it is defined as "fold-change of tracer-bound signal (present at the recommended concentration) compared to the concentration of the free protein", however the figures show the assay window defined as 50% and 100% of max signal. We recommend that the authors modify this prior to publication, as we anticipate this may be confusing for users, especially to those who are not experts in assay design.

Reviewer #3 (Remarks to the Author):

I would like to thank the authors for providing a revised manuscript that satisfactorily addresses all my comments.

Reviewer #2 (Remarks to the Author):

The authors have made significant efforts to address the concerns of the previous review, and the authors' references to following the format of the Chemical Probes Portal provides confidence that the database will be appropriately managed following increased user engagement. Additionally, the authors' statement regarding publishing in a less specialised journal (instead of a chemistry or biology-focussed journal) to gain access to a wider audience is valid. We would thus recommend for publication in Nature Communications following minor revisions.

Minor points

The concern regarding assay window definition has not been adequately addressed, with differing/conflicting definitions on tracerDB and the manuscript. In the manuscript the term 'assay window' is not defined and the figure shows 50% max signal, whilst on the tracerDB 'about' section it is defined as "fold-change of tracer-bound signal (present at the recommended concentration) compared to the concentration of the free protein", however the figures show the assay window defined as 50% and 100% of max signal. We recommend that the authors modify this prior to publication, as we anticipate this may be confusing for users, especially to those who are not experts in assay design.

We thank the reviewer for the support and the valuable comments. We have already received positive feedback from users about the updated database with improved usability. We are confident that this publication will be helpful in reaching a wide range of scientists and increasing participation, and we greatly appreciate the input we have received.

We also agree with the minor revision regarding the assay window. We have added a brief explanation in the manuscript and in the figure captions on the 'About' page to clarify the dependence of the assay window on the recommended concentration. We hope that this now sufficiently describes the assay window parameter that is reported in the database.

Reviewer #3 (Remarks to the Author):

I would like to thank the authors for providing a revised manuscript that satisfactorily addresses all my comments.

We thank the reviewer for the valuable feedback which allowed us to notably increase the database's quality and we are happy that we addressed all comments.